# Characterization Data for the Establishment of Scale-Up and Process Transfer Strategies between Stainless Steel and Single-Use Bioreactors

**Vincent Bernemann** [1,*], **Jürgen Fitschen** [2], **Marco Leupold** [3], **Karl-Heinz Scheibenbogen** [3], **Marc Maly** [1], **Marko Hoffmann** [1], **Thomas Wucherpfennig** [2] and **Michael Schlüter** [1,*]

1   Institute of Multiphase Flows, Hamburg University of Technology, 21073 Hamburg, Germany; marc.maly@tuhh.de (M.M.); marko.hoffmann@tuhh.de (M.H.)
2   Boehringer Ingelheim Pharma GmbH & Co. KG, 88400 Biberach, Germany; juergen.fitschen@boehringer-ingelheim.com (J.F.); thomas.wucherpfennig@boehringer-ingelheim.com (T.W.)
3   Sartorius Stedim Biotech, 37079 Göttingen, Germany; marco.leupold@sartorius.com (M.L.); karl-heinz.scheibenbogen@sartorius.com (K.-H.S.)
*   Correspondence: vincent.bernemann@tuhh.de (V.B.); michael.schlueter@tuhh.de (M.S.)

**Abstract:** The reliable transfer of bioprocesses from single-use bioreactors (SUBs) of different scales to conventional stainless steel stirred-tank bioreactors is of steadily growing interest. In this publication, a scale-up study for SUBs with volumes of 200 L and 2000 L and the transfer to an industrial-scale conventional stainless steel stirred-tank bioreactor with a volume of 15,000 L is presented. The scale-up and transfer are based on a comparison of mixing times and the modeling of volumetric mass transfer coefficients $k_La$, measured in all three reactors in aqueous PBS/Kolliphor solution. The mass transfer coefficients are compared with the widely used correlation of van't Riet at constant stirrer tip speeds. It can be shown that a van't Riet correlation enables a robust and reliable prediction of mass transfer coefficients on each scale for a wide range of stirrer tip speeds and aeration rates. The process transfer from single-use bioreactors to conventional stainless steel stirred-tank bioreactors is proven to be uncritical concerning mass transfer performance. This provides higher flexibility with respect to bioreactor equipment considered for specific processes.

**Keywords:** scale up; single-use bioreactor (SUB); mass transfer; mixing time; stirred-tank reactor (STR); $k_La$; correlation; PBS; Kolliphor

## 1. Introduction

Biopharmaceutical processes, like mammalian cell culture or microbial processes, have gained more and more relevance on the market over recent decades. With this ongoing development, the performance of bioreactors comes into focus for increasing the productivity and quality of the products. To obtain those goals, single-use systems are increasingly being used, especially at the small and medium scale [1]. The term "single-use" describes a bioreactor whose cultivation container is only used one single time and therefore is made of sterile FDA-approved disposable plastics. As conventional stainless steel bioreactors have been used for a large variety of different products in the past, the question of if and how those processes can be transferred and adapted to single-use systems is of increasing importance. In the literature, the specific power input, the mass transfer performance, and mixing time are often described as characterization parameters [2,3]. However, when transferring processes between different tank designs as well as between stainless steel and single-use bioreactors, the processes cannot be transferred directly. This is primarily attributed to the differing geometry of the reactors. To address this issue, this work aims to provide a scale-up strategy between single-use bioreactors (Sartorius Biostat STR® 200 L and 2000 L) and the process transfer to conventional stainless steel stirred-tank bioreactors (15,000 L).

## 2. Materials and Methods

The scale-up and scale-down of bioprocesses in bioreactors try to deliver similar performances between scales, while quantifiable characteristics are often highly dependent on the overall bioreactor's geometry and the volume used. Several factors like $P/V$, $k_L a$, tip speed, and aeration rate can be considered separately or in combination. In this study, the volumetric mass transfer coefficient $k_L a$ and the mixing time are in focus to elaborate a scale-up directly over the process parameters rather than an indirect scale-up over operating conditions.

The $O_2$ mass transfer performance is a key parameter in cell cultivation as it is crucial for cell growth and metabolism. Therefore, detailed knowledge of the oxygen transfer rate (OTR) [4], depending on the liquid-side mass transfer coefficient $k_L$, the volumetric surface area $a$, and the difference between the saturation concentration $c_{O2}^*$ and the average dissolved oxygen concentration in the liquid phase $c_{02}$ [5], is of high importance. As for both the mass transfer coefficient $k_L$ and the volumetric surface area $a$, individual measurements are challenging. It is best practice in bioprocess engineering to measure the product of both, called the volumetric mass transfer coefficient $k_L a$. During recent decades, a lot of research has been conducted and published for bioreactors concerning the volumetric mass transfer coefficient $k_L a$ in terms of experimental methods, modelling with empirical correlations, and numerical simulations [1,6]. Nevertheless, the transfer of processes from one scale to another and recently from conventional stainless steel stirred-tank bioreactors to single-use bioreactors and vice versa can be challenging [7,8].

Furthermore, knowledge about the mixing time performance is of great importance for the reliable supply of nutrients and for the prevention of concentration gradients and should always be considered during scale-up. Particularly in fed batch cultivations, the mixing performance can significantly change depending on the filling volume, the specific power input, and the aeration rate [2,8].

Therefore, experimental results for the characterization of the mixing time and oxygen mass transfer performance for different bioreactors are shown, compared, and discussed in this work.

### 2.1. Reactor Setup and Operation Parameters

The three different reactors that are investigated and compared are an acrylic glass replica of a stainless steel 15,000 L production vessel, an acrylic glass replica of a Sartorius Biostat STR® 2000 L, and an acrylic glass replica of a Sartorius Biostat STR® 200 L. As the advantage of acrylic glass replicas is their optical accessibility, the use of non-transparent Sartorius Flexsafe STR® bags was not needed. All bioreactors considered were designed as stirred tanks and showed geometrical similarities, whereas the 15,000 L system, in contrast to the Biostat STR family, features some differences. In contrast to the smaller SUBs, the stainless steel replica contains four baffles, and has different impeller types and a different shape concerning the bottom geometry and aspect ratio. The geometrical parameters and operation conditions are summarized in Table 1 and the setups are shown in Figure 1. All reactor replicas used have been qualified to match the performance of the real reactor systems at Sartorius and Boehringer Ingelheim. The operating conditions used are based on industrially relevant parameters and have been selected in coordination with the project partners. The volume flow rates are given in standard liters per time.

**Table 1.** Characteristics and operation parameters of the used systems.

| System | Stainless Steel | Biostat STR® 2000 L | Biostat STR® 200 L |
|---|---|---|---|
| Working Volume | 8000 L–12,500 L | 1400 L–2000 L | 160 L–200 L |
| Vessel Diameter | 2.000 m | 1.295 m | 0.595 m |
| Impeller Type (bottom, top) | Rushton, Pitched-Blade | Rushton, Segment | Rushton, Segment |
| Impeller Diameter | 0.665 m | 0.492 m | 0.225 m |

**Table 1.** *Cont.*

| System | Stainless Steel | Biostat STR® 2000 L | Biostat STR® 200 L |
|---|---|---|---|
| Angle Top Impeller | 45° | 30° | 30° |
| Stirrer Speed Range | 30 rpm–80 rpm | 35 rpm–70 rpm | 62 rpm–150 rpm |
| Aeration Range | 20 Lpm–250 Lpm | 10 Lpm–100 Lpm | 1 Lpm–20 Lpm |
| Sparger Type | Open Tube | Ring (0.8 mm × 200) | Ring (0.8 mm × 20) |
| Baffles | 4 × 200 mm | no baffles | no baffles |

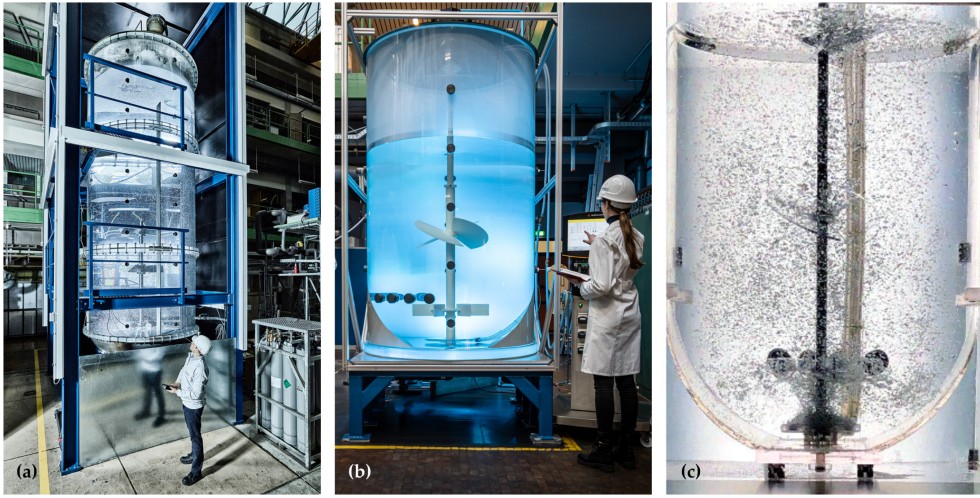

**Figure 1.** Images of the different reactors used: (**a**) Acrylic glass replica of a 15,000 L stainless steel bioreactor from Boehringer Ingelheim. (**b**) Acrylic glass replica of a Biostat STR® 2000 by Sartorius. (**c**) Acrylic glass replica of a Biostat STR® 200 by Sartorius.

### 2.2. Experimental Procedure to Measure the Global Mixing Time

The global mixing time of the stirred tanks investigated in this work is based on the optical decolorization method of a pH-sensitive tracer utilizing the advantage of the fully transparent reactor design. The optical decolorization method has the advantage over local measurement techniques that potential dead zones and poorly mixed areas can be identified and described [9]. For this purpose, the course of the decolorization of bromothymol blue over time is analyzed using a chemical neutralization reaction. The definition of the mixing time is based on the 95% criterion [10–12]. For the neutralization reaction, a 50 mmol/L bromothymol blue–ethanol solution is initially used to establish a 3 µmol/L bromothymol blue concentration in the respective reactors. Subsequently, the pH value is raised into the basic range using a 2 M NaOH solution to color the reactor dark blue. A volume corresponding to 0.02‰ of the reactor volume is added. To ensure good mixing of the NaOH in the reactor volume, stirring is carried out for at least 5 min at the operating parameters to be investigated. Subsequently, 0.04‰ of the reactor volume of 2 M HCl solution is added to the surface and the course of the decolorization is recorded with a Nikon D7500. A detailed description of the method and the evaluation can be found in Fitschen et al. [11].

### 2.3. Experimental Procedure to Measure the Volumetric Mass Transfer Coefficient

As the measurement method the gassing-out method without organisms is used, according to the DECHEMA guideline [4], as it is a cost-effective, well-described, and widely used method [4,5,13]. In the first step, the oxygen concentration in the liquid is stripped down to below 20% a.s. using nitrogen aeration. In the second step, the desired stirrer frequency is set, and the system is aerated with pressurized air at different gas flow rates until a dissolved oxygen concentration of $c_{O2} > 80\%$ a.s. is reached. In contrast to the DECHEMA guideline, no headspace exchange was performed, as the reactors were not

covered by a lid, and additional headspace aeration showed no influence on the results. The oxygen concentration was measured using an FDO®925 probe from WTW, Weilheim, Germany. The time constants of the probes were measured to $t_e = 12.5$ s, indicating a suitable probe response time for the measured values, as the error is estimated to be below 5% for the highest $k_L a$ [14,15]. The oxygen saturation concentration $c_{O2}^*$ was measured in each vessel prior to the experiments to avoid an influence of the water column. All experiments were performed at 37 °C in an aqueous solution with $1\times$ PBS (phosphate-buffered saline) according to DECHEMA [4] with additional 1 g/L Kolliphor® from BASF SE, Ludwigshafen, Germany. The use of an additional 1 g/L of Kolliphor as a polaxomer in the characterization is important, since Kolliphor has a significant influence on the surface tension and on the mass transfer [16]. The combination of PBS and 1 g/L Kolliphor is thus a good model medium which represents the rheological properties of cell culture media. Assuming the applicability of the two-film theory, the oxygen transfer rate

$$OTR = \frac{dc_{O2}}{dt} = k_L a \cdot (c_{O2}^* - c_{O2}),\tag{1}$$

depends on the volumetric mass transfer coefficient $k_L a$, the saturation concentration $c_{O2}^*$, and the transient dissolved oxygen concentration $c_{O2}(t)$.

*2.4. Modeling of Mass Transfer Performance*

One of the most frequently used correlations to predict the volumetric mass transfer performance in bioreactors is the one introduced by van't Riet [13]:

$$k_L a = C \cdot \left(\frac{P}{V}\right)^\alpha \cdot v_S^\beta,\tag{2}$$

where the volumetric mass transfer coefficient $k_L a$ is correlated via the specific power input $P/V$ and the superficial gas velocity $v_S$, weighted with the empirical parameters $C$, $\alpha$, and $\beta$.

With different filling volumes used for each reactor type, a fourth dependency is proposed representing the volume itself to optimize the fit of the function. Especially for reactor scales $V > 1000$ L, the reactor height has a strong influence on the mass transfer as the gas residence time increases with higher water columns.

Further, the aeration rate is represented by the volumetric aeration rate *vvm*, as it is a widely used parameter in the bioprocess industry and represents the volume exchange by aeration within the reactor. The volumetric power input is changed to the stirrer tip speed $u_{tip}$, which is a more easily accessible entity compared to the power input. These adaptions lead to a modified van' Riet correlation

$$k_L a_{mod} = C \cdot \left(\frac{u_{tip}}{m \cdot s^{-1}}\right)^\alpha \cdot \left(\frac{vvm}{min^{-1}}\right)^\beta \cdot \left(\frac{V}{m^3}\right)^\gamma,\tag{3}$$

using four model parameters. For this correlation, the units are chosen as $[u_{tip}] = \frac{m}{s}$, $[vvm] = \frac{L}{min \cdot L}$, and $[V] = m^3$.

## 3. Results

All experiments were carried out in the acrylic glass replicas at the Institute of Multiphase Flows at Hamburg University of Technology. In the following, the experimental results on mixing time and mass transfer performance will be presented and discussed.

*3.1. Mixing Time*

In Figure 2, the results of the global mixing time in relation to the stirrer tip speed are presented for the different reactors. The results of the mixing time without aeration are represented as empty symbols and those of the mixing time with aeration as filled symbols.

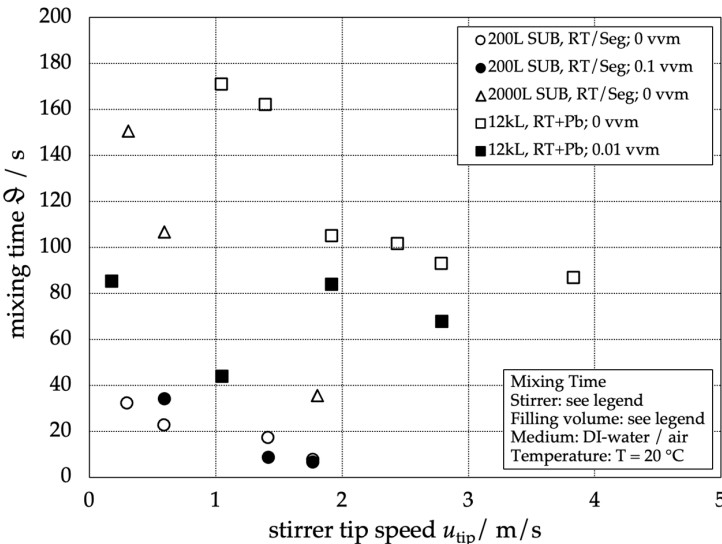

**Figure 2.** Visualization of the global mixing time for different stirred-tank reactors and aeration rates depending on the stirrer tip speed. Working volumes are given in the legend.

As expected, the global mixing time for single-phase operation increases with increasing reactor scale and decreases with increasing stirrer tip speed. For the aerated conditions, however, no clear trend is recognizable. Its absence can be explained by the dominant influence of the buoyancy-driven flow caused by the rising gas bubbles leading to heterogeneous flow patterns, as already reported in [6]. The scaling for the unaerated systems can be simplified by plotting the dimensionless mixing time

$$\theta = \vartheta \cdot n,\tag{4}$$

calculated from the mixing time $\vartheta$ and the stirring frequency $n$, in dependency of the Reynolds number (Figure 3)

$$Re = \frac{\rho_L \cdot n \cdot d_R}{\eta_L},\tag{5}$$

calculated with the density of the liquid $\rho_L$, the impeller diameter $d_R$ and the dynamic viscosity of the liquid $\eta_L$.

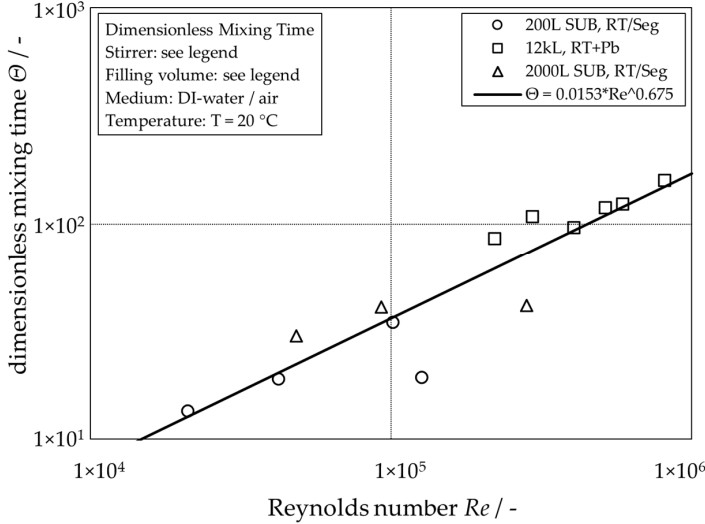

**Figure 3.** Visualization of the global dimensionless mixing time of different unaerated stirred-tank reactors depending on the stirrer Reynolds number.

The line of best fit for the single-phase dimensionless mixing times represents a good approximation of the data points over a wide range of Reynolds numbers for all three reactor scales. Figure 2 shows that mixing times with additional gassing can be expected to be faster than the single-phase mixing time [9].

### 3.2. Mass Transfer Performance and Modelling

As a second important parameter for the biopharmaceutical industry, the volumetric mass transfer coefficient $k_L a$ is determined as described in Section 2.3. For each reactor, 22 measurements in the parameter range according to Table 2 are performed. All measured data can be found in the Data Availability Statement. Using these data as input, the coefficients *C*, $\alpha$, $\beta$, and $\gamma$ are determined individually by using the 22 measured data points for each type of reactor as input parameters for a MATLAB 2021b script (provided via GitLab, see Data Availability Statement). The script uses a least square optimization function. Figure 4 shows an example of the fitting of the extended van't-Riet Equation (3) with the above-mentioned set of parameters. The individual coefficients for the three different types of reactors are shown in Table 3, and their validity range (range for which the parameters have been proven) is noted in Table 2.

**Table 2.** Ranges of validity for the used correlation in Equation (3) from data-gathering experiments.

| | Stainless Steel Stirred Tank | Biostat STR® 2000 L | Biostat STR® 200 L |
|---|---|---|---|
| $u_{tip}$ | $\in \left[1.04\frac{m}{s}, 2.79\frac{m}{s}\right]$ | $\in \left[0.90\frac{m}{s}, 1.80\frac{m}{s}\right]$ | $\in \left[0.73\frac{m}{s}, 1.77\frac{m}{s}\right]$ |
| *vvm* | $\in \left[0.0017\ \text{min}^{-1}, 0.0208\ \text{min}^{-1}\right]$ | $\in \left[0.0050\ \text{min}^{-1}, 0.0500\ \text{min}^{-1}\right]$ | $\in \left[0.0050\ \text{min}^{-1}, 0.1000\ \text{min}^{-1}\right]$ |
| *V* | $\in \left[8\ \text{m}^3, 12.5\ \text{m}^3\right]$ | $\in \left[1.4\ \text{m}^3, 2.1\ \text{m}^3\right]$ | $\in \left[0.16\ \text{m}^3, 0.2\ \text{m}^3\right]$ |

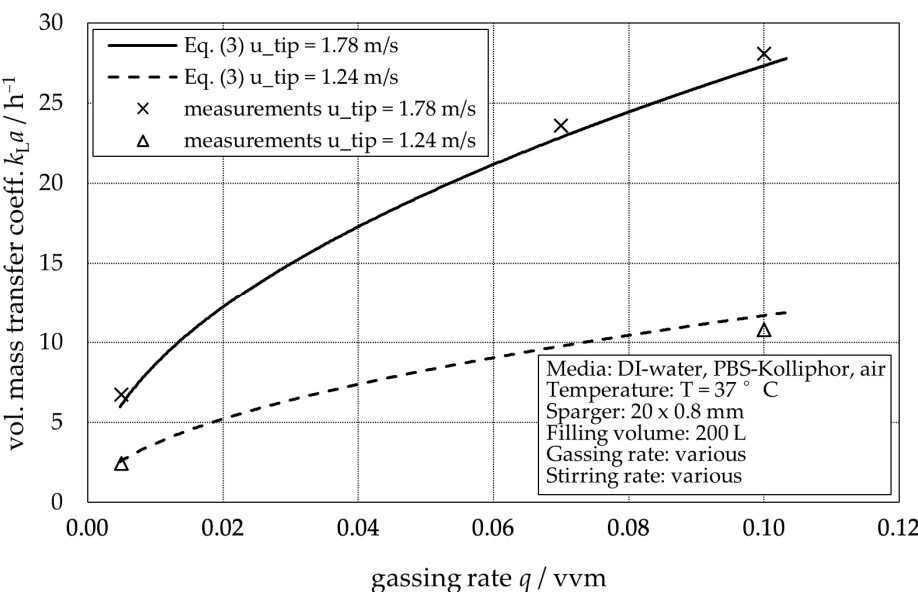

**Figure 4.** Example of measured $k_L a$ data compared to correlation (Equation (3)) with parameters from Table 3. Exemplarily shown for STR® 200 L, $u_{tip} \in \{1.24, 1.78\}$. Correlation data are based on the whole dataset of 22 points, not only the depicted examples.

The parity plots of the predicted and measured $k_L a$ values for the three different types of reactors show that the deviation of Equation (3) is smaller than 15% in almost the whole range of parameters (Figure 5). It must be emphasized that the set of parameters is kept constant for each reactor for the full range of operation conditions (see Table 3). The correlation enables accurate prediction for all types of reactors and a comparison between the reactors based on typical design criteria, like stirrer tip speed $u_{tip}$.

**Table 3.** Coefficients for $k_\mathrm{L}a$ estimation in each reactor type with PBS-Kolliphor solution at 37 °C according to Equation (3).

| Coefficient | Stainless Steel Stirred Tank | Biostat STR® 2000 L | Biostat STR® 200 L |
|---|---|---|---|
| C | 25.84 | 29.40 | 25.58 |
| α | 1.64 | 1.53 | 2.45 |
| β | 0.92 | 0.69 | 0.50 |
| γ | 0.81 | 0.55 | 0.11 |

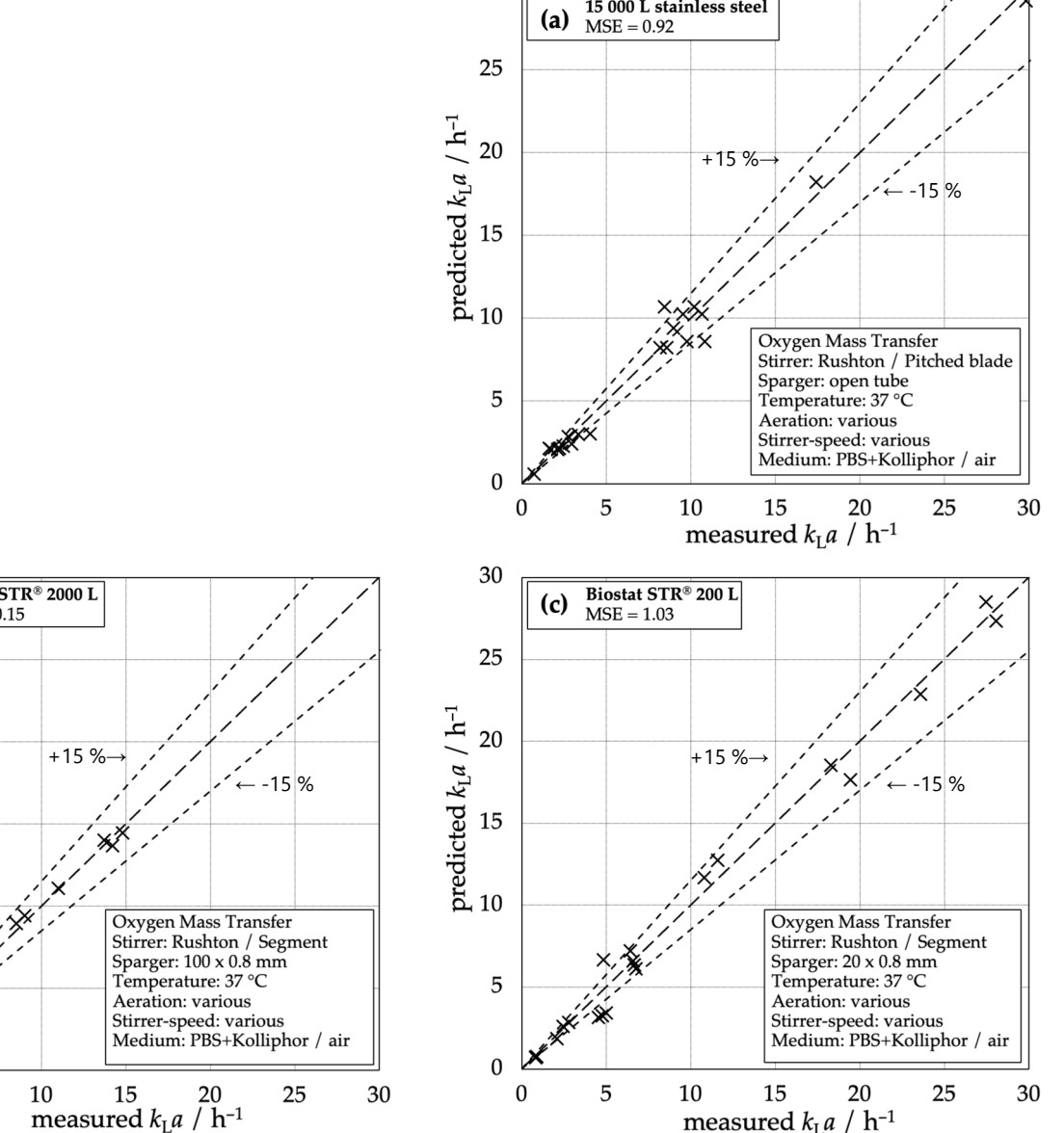

**Figure 5.** Measured $k_\mathrm{L}a$ compared to the predicted $k_\mathrm{L}a$ correlated with Equation (3) and coefficients from Table 3: (**a**) 15,000 L stainless steel reactor, (**b**) Biostat STR® 2000 L, (**c**) Biostat STR® 200 L.

In addition, by comparing the parameters, some insights into the physical processes and their influence on mass transfer performance can be gained.

According to Equation (3), the coefficient β reflects the influence of the volumetric aeration rate *vvm*, which increases with the reactor volume. The 15,000 L vessel shows the highest value for coefficient β, whereas the 200 L shows the lowest. A reason for

this is the residence time of the gas inside the reactor. With the increasing height of the reactor, the residence time increases, and more oxygen is transferred into the liquid phase. Therefore, the usage of the gaseous phase In the smaller scales is not as efficient as in the higher reactors. The same effect leads to an increasing volume coefficient $\gamma$ with greater reactor scales.

The coefficient $\alpha$ reflects the influence of the stirrer tip speed ($u_{\text{tip}}$), which is related to the agitation rate and stirrer diameter. Table 3 shows the highest impact of the agitation rate at the 200 L scale. This is related to the fact that the aeration rate has a comparatively lower impact at smaller scales, as discussed above, leading to a more significant impact of the stirrer frequency on the mass transfer performance, as the presence of small bubbles with larger interfacial area $a$ in the system is more valuable. The $\alpha$ coefficient for both the STR® 2000 L and the 15,000 L systems is significantly lower compared to that of the STR® 200 L. Despite having a lower reactor height, $\alpha$ for the STR® 2000 L is lower compared to the 15,000 L system. This is due to the different reactor geometries, which include baffles in the 15,000 L reactor. The baffles prevent the formation of a tangential flow and lead to a higher efficiency of the impellers. Furthermore, at a constant tip speed, with a larger impeller diameter, bigger and more efficient trailing vortices are created behind the stirrer blades, which enhance the dispersion of the gas bubbles [9].

The influence of the reactor filling volume increases with the reactor scale, reflected by the coefficient $\gamma$. With the given reactor diameter, the filling volume is proportional to the used water column, favoring larger scales for the mass transfer performance.

In summary, it can be concluded that the measured volumetric mass transfer coefficients for all three reactors are all within the same order of magnitude and can be predicted well by using the modified van't Riet correlation (3) with a single set of parameters for each type of reactor.

### 3.3. Comparison of the Correlations for Different Scales and Reactors

To compare the different reactor types and scales, the previously determined correlations are used. By comparing the correlations instead of discrete data points, it is possible to compare continuous graphs and the influence of different parameters.

Figure 6 shows the oxygen mass transfer performance of the 15,000 L stainless steel bioreactor compared to the STR® 200 L single-use bioreactor based on the same stirrer tip speeds as an important design and scale-up criterion. With increasing tip speed, the volumetric mass transfer coefficients $k_{\text{L}}a$ are increasing in both types of reactors. In this figure, the graphs of two exemplarily aeration rates (0.005 vvm and 0.01 vvm) are shown. The graph shows that the predicted mass transfer coefficient $k_{\text{L}}a$ in the 200 L SUB increases faster with increasing tip speed, compared to the 15,000 L STR. This means that at the example aeration rate of 0.005 vvm, a higher tip speed is necessary for the 15,000 L stainless steel reactor to reach the same $k_{\text{L}}a$ values as the STR® 200 L. With increasing volumetric mass transfer coefficients (resulting from higher tip speeds), the divergence increases. Using a higher aeration rate of 0.01 vvm, the performance of the 15,000 L stainless steel reactor is higher at low tip speeds because the dispersion of the large-scale Rushton Turbine is more efficient. A volumetric mass transfer coefficient $k_{\text{L}}a = 5.0 \text{ h}^{-1}$ is achieved at approximately the same stirrer tip speed.

The higher the aeration rate, the more the performance of the stainless steel reactor equals the performance of the smaller system, as the height of the reactor and the gas residence time gain importance. This shows that a process transfer from SUBs to the 15,000 L stainless steel reactors should be of low risk concerning mass transfer performance and widens the design space for the SUBs.

Comparing the single-use bioreactors STR® 2000 L and STR® 200 L using the stirrer tip speed as a scaling parameter, Figure 7 shows a higher mass transfer performance for the STR® 200 L. With higher gas flow rates, the difference between both reactor scales decreases. This can be explained by the different flow pattern in the unbaffled system. In the 200 L SUB, the tubing and specially shaped bottom lead to stronger turbulence and bubble

dispersion compared to the 2000 L SUB where a vortex is formed with less disturbances. The larger reactor diameter leads to a higher angular momentum and, in relation to the reactor diameter, the smaller tubing in the overall larger STR 2000 L SUB causes a smaller baffling effect, leading to a more pronounced tangential flow and thus vortex.

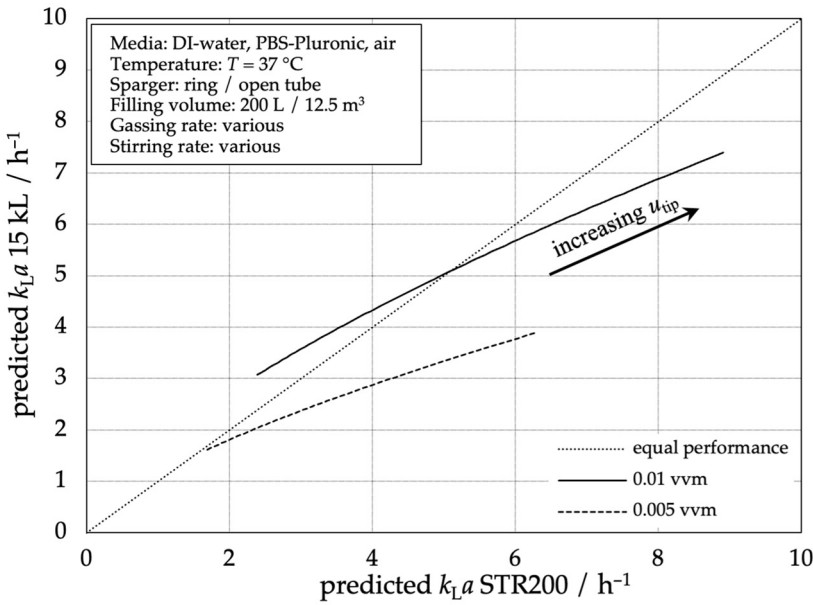

**Figure 6.** Comparison of the predicted mass transfer performance according to Equation (3) of the 15,000 L stainless steel reactor compared to the STR® 200 L.

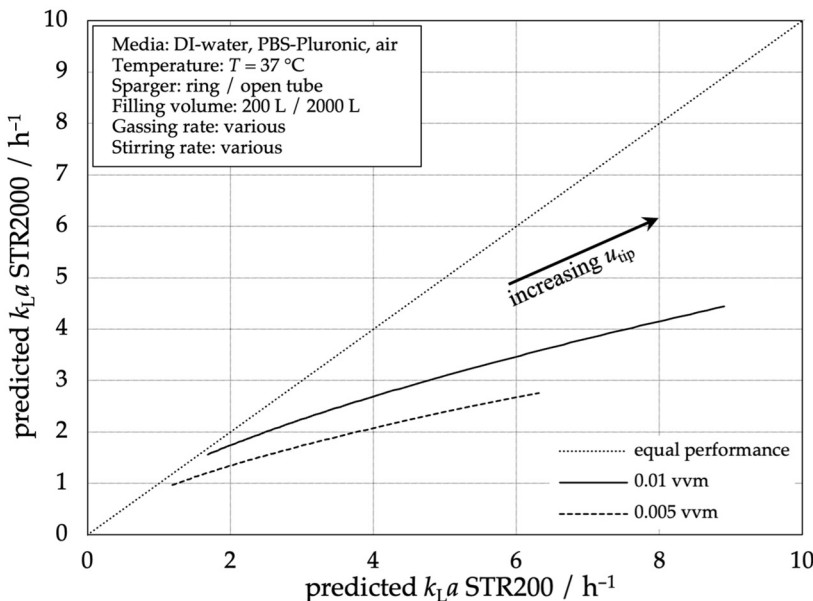

**Figure 7.** Comparison of predicted mass transfer performance according to Equation (3) in the STR® 200 L compared to the predicted mass transfer performance of the STR® 2000 L.

Figure 8 shows the performance of the 15,000 L stainless steel reactor compared to the STR® 2000 L. For all data points, the stainless steel reactor shows a higher mass transfer performance with even higher efficiency at higher gas flow rates. The graph shows a linear dependency between both reactor types, meaning that the stainless steel's performance is proportional to the STR® 2000 L's performance with an aeration-dependent factor.

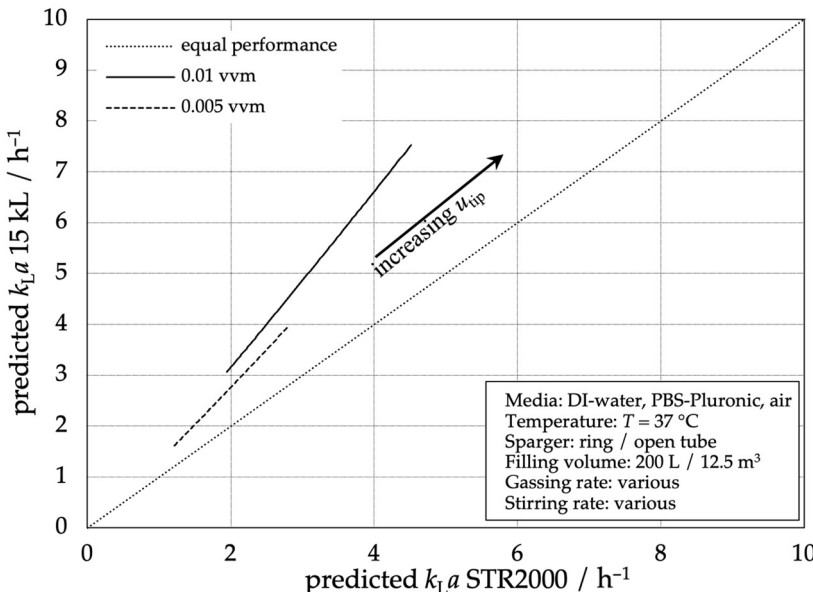

**Figure 8.** Comparison of the predicted mass transfer performance according to Equation (3) of the 15,000 L stainless steel reactor compared to the STR® 2000 L.

Once again, the possibility of a reliable process transfer from the SUB to 15,000 L scales concerning mass transfer performance is evident. For a transfer from the 15,000 L scale to the 2000 L SUB, additional aeration should be considered to match the volumetric mass transfer performance.

## 4. Discussion

In the past, the van't Riet correlation has been widely used to describe the volumetric mass transfer coefficient in conventional bioreactors. Figure 5 shows that the chosen modified van't Riet correlation (Equation (3)) is applicable to conventional as well as single-use systems with an acceptable accuracy, as most of the data points lay within a range of ±15%. The fit becomes more precise for higher $k_L a$ values in all systems. The correlation is best applicable for the STR® 2000 L system with a calculated mean square error of MSE = 0.14 and no estimated data point varying more than 15% from the measured ones. Even for the stainless steel reactor and the STR® 200 L, the correlation is useful as the few single outliers do not deviate strongly from 15% and only a few input parameters are necessary to estimate the mass transfer performance. The correlation is very handy to compare different systems and scales, even if the single-use systems have a different flow pattern compared to the conventional reactor type due to different stirring and gassing devices and the lack of baffles. The applicability of the correlation over all reactor scales and types lays the foundation for the comparison of the reactors.

Furthermore, it is clear that for high aeration rates the large-scale systems perform better than small-scale systems because of the higher gas residence time in larger dimensions, making use of the higher water column, especially in systems with higher aspect ratios. The graphs also show that the smallest scale is most sensitive to the stirrer tip speed, whereas the gassing rate drives the performance of higher systems.

At low aeration rates, the performance of the STR® 2000 L is comparable to that of the STR® 200 L. However, with increasing aeration rate the STR® 200 L gains in mass transfer performance. To achieve equal performance in the STR® 2000 L, the stirrer tip speed or the aeration rate must be increased. Also, the micro gassing device, which uses more and smaller holes, can be used to increase the mass transfer performance.

Comparing the STR® 2000 L with the conventional stainless steel reactor, Figure 8 shows the possibility of a scale-up between both reactor types based on the mass transfer coefficient. This implies the possibility of lower stirrer tip speeds or lower aeration rates

on a larger scale, which could be important for upstream processing or the transfer of products from single-use bioreactors to conventional large-scale stirred-tank bioreactors. In the different systems with the various experiments, different flow patterns of the gas occur in which the gas is often not distributed evenly (especially in the SUBs). However, a scale-up still is possible between stainless steel and single-use reactors.

As a second parameter, the dimensionless mixing time for the unaerated systems is useful as a scale-up criterion between single-use and conventional systems. However, the mixing time under aeration is complex to predict over different scales and reactor types as the different geometries and spargers induce different flow patterns, causing diverging mixing times. Assuming that mixing time with aeration is faster than without aeration, the proposed scale-up approach for the single-phase mixing time can be used as a worst-case approximation [9]. A transfer of the results to real cultivation processes can be carried out assuming that the rheological properties of the fluids do not change significantly.

## 5. Conclusions

It has been shown that the widely used correlation introduced by van't Riet [13] for estimating the mass transfer capacity for stirred-tank reactors can be applied with a small modification for both conventional stirred-tank reactors as well as single-use bioreactors of different scales. This enables the use of the proposed correlation as a tool for the scale-up or scale-down of single-use reactors as well as process transfer to stainless steel bioreactors. A comparison between single-use bioreactors on two different scales and an industrial scale conventional stainless steel bioreactor shows the good applicability of the proposed correlation for the prediction of the oxygen mass transfer performance. In combination with the prediction of dimensionless mixing times, reliable scaling and process transfer are possible. As the scope of this work is only related to Biostat STR® reactors with one single stirrer geometry and a ring sparger, it must be noted that higher mass transfer performances can be expected with the available micro sparger with 0.15 mm hole diameter and Segment/Segment impeller geometry. This will result in different correlation coefficients and therefore provide another parameter set for scale-up and process transfer, based on the specific needs and criteria. Related to these parameters and in context to interconnected scaling parameters, additional information about the scale-up and scale-down for the Sartorius Biostat STR® reactors can be extracted from the BioPAT® Process Insights Tool provided by Sartorius.

To ensure constant process conditions regarding the mixing time and mass transfer performance for cell culture applications over all reactor scales, detailed knowledge of the individual reactor performance is essential. In this work, it is shown that a reliable process transfer between the investigated systems is possible without loss in performance and with an easy-to-use first-hand approach.

The extension of the validity range, the transfer to other bioreactor scales, geometries, and setups, as well as the improvement in reactor designs are subjects of future research to further increase the speed and safety of process transfers in the biopharmaceutical industry.

**Author Contributions:** Conceptualization, J.F., T.W. and M.S.; methodology, J.F.; software, V.B.; validation, J.F. and V.B.; data curation, V.B.; writing—original draft preparation, V.B.; writing—review and editing, J.F., M.L., K.-H.S., T.W., M.M., M.H. and M.S.; visualization, V.B.; supervision, M.M. and M.S.; project administration, M.S. and M.L.; funding acquisition, M.H. All authors have read and agreed to the published version of the manuscript.

**Funding:** This research received no external funding.

**Data Availability Statement:** All individual data points used for the article as well as the MATLAB script for determination of coefficients in Table 3 can be found at https://collaborating.tuhh.de/v5/multiphase-bioreactors/vincent-bernemann/mdpi-fluids-scale-up.

**Acknowledgments:** The authors gratefully thank Philipp Hebestreit from BASF SE for the support with Kolliphor P188 Bio to conduct the presented research. The authors gratefully thank Boehringer Ingelheim Pharma GmbH & Co. KG and Sartorius Stedim Biotech for the financial support. Publish-

**Conflicts of Interest:** Authors Jürgen Fitschen and Thomas Wucherpfennig were employed by the company Boehringer Ingelheim Pharma GmbH & Co. KG and authors Marco Leupold and Karl-Heinz Scheibenbogen were employed by the company Sartorius Stedim Biotech. All authors declare that the research was conducted in the absence of any commercial or financial relationships that could be construed as a potential conflict of interest.

## Abbreviations

The following abbreviations are used in this manuscript:

| | |
|---|---|
| DECHEMA | Gesellschaft für Chemische Technik und Biotechnologie e. V. |
| Eq | equation |
| FDA | Food and Drug Administration |
| MSE | mean squared error |
| OTR | oxygen transfer rate |
| Pb | pitched blade impeller |
| PBS | phosphate-buffered saline |
| RT | Rushton Turbine |
| Seg | segment impeller |
| STR | stirred-tank reactor |
| SUB | single-use bioreactor |
| TUHH | Hamburg University of Technology |

**Nomenclature**

Characteristic Numbers

| | | |
|---|---|---|
| $Re$ | stirrer Reynolds number | $Re = \frac{\rho_L \cdot n \cdot d_R}{\eta_L}$ |

Greek Symbols

| | | |
|---|---|---|
| $\alpha$ | empirical parameter | - |
| $\beta$ | empirical parameter | - |
| $\gamma$ | empirical parameter | - |
| $\vartheta$ | mixing time | s |
| $\theta$ | dimensionless mixing time | - |
| $\rho$ | density | $\mathrm{kg\,m^{-3}}$ |
| $\eta$ | dynamic viscosity | Pa s |

Roman Symbols

| | | |
|---|---|---|
| $a$ | specific area | $\mathrm{m^{-1}}$ |
| $c$ | concentration | - |
| $C$ | empirical parameter | - |
| $c^*$ | saturation concentration | - |
| $d$ | diameter | m |
| $k$ | mass transfer coefficient | $\mathrm{m\,s^{-1}}$ |
| $n$ | stirring frequency | $\mathrm{s^{-1}}$ |
| $OTR$ | oxygen transfer rate | $\mathrm{s^{-1}}$ |
| $P$ | power | W |
| $t$ | time | s |
| $T$ | temperature | °C |
| $u$ | stirrer velocity | $\mathrm{m\,s^{-1}}$ |
| $V$ | volume | $\mathrm{m^3}$ |
| $v$ | gas velocity | $\mathrm{m\,s^{-1}}$ |
| $vvm$ | volumetric gassing rate | $\mathrm{s^{-1}}$ |

Subscripts

| | |
|---|---|
| e | electrode |
| L | liquid |
| mod. | modified |

| | |
|---|---|
| O$_2$ | oxygen |
| R | impeller |
| S | superficial |
| tip | stirrer tip |

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
