# Peer review of "Characterization Data for the Establishment of Scale-Up and Process Transfer Strategies between Stainless Steel and Single-Use Bioreactors"

_fluids, doi:10.3390/fluids9050115_

Round 1

Reviewer 1 Report

Comments and Suggestions for Authors

Dear Authors,

Your manuscript was interesting to read. Here are my feedback and suggestions for improving your article.

Abstract & Introduction - The application of P/V criterion for scaling up may be simpler than kLa. Could you elucidate the advantages of your methodology as compared to P/V method (or P/V plus vvm) for scaling up across Single Use Bioreactors (SUBs) and the stainless steel one with geometrical dissimilarities?

Table 1 – please clarify the criteria based on which you have determined the operating range (within the design space) for stirrer speed and aeration in the absence of active culture in your gassing out method?

Materials & Methods - How back pressure of the 15K stainless steel bioreactor differed from the SUBs? Additionally, please discuss the potential impact of this parameter, and how it should be taken into account for the estimation of kLa across different bioreactor types (SUBs and stainless steel one). 

Materials & Methods - How did you calculate the saturation concentration of O2 (ppm, mg/L), in your gassing out method in order to estimate the actual kLa value?

Materials & Methods - Could you briefly elaborate on the rationale(s) behind selecting a combination of kLa and mixing time as the scaling strategy as compared to other strategies? 

Materials & Methods – please make sure if the WTW FDO 925 probe you have used is suitable for the experiment in terms of response time.

Results - Please ensure that all figures and figure legends adhere to the requirements specified by the journal.

Figure 2 - Please verify that the 2000L SUB, RT at 0.01 vvm is included in the figure.

Line 158 from However to end of Line 160 – please paraphrase your sentences.

Line 161 & 333 – please provide brief explanation on how you calculated/or predicted the dimensionless mixing time and perhaps characteristic time scale? 

Results - please include a brief explanation for each element of the Reynolds number equation, detailing what they refer to and how they has been calculated, where applicable.

Line 169 from "Figure 2 … " should be transferred to the line 160-161. 

Line 221 - Starting from "Furthermore...", please make sure your statement regarding the tested conditions is accurate.

Line 270 - Please briefly elaborate on what factors contribute to the generation of a vortex in the 2000 L SUB and discuss potential methods for its prevention.

Discussion- please briefly elaborate on major strategies to improve the kLa performance in SUB of smaller size, particularly addressing challenges arising from the absence of back pressure and baffles.

Discussion - Please incorporate your reasoning behind why you believe the scale-up strategy you have employed (kLa & mixing time) can be reliably applied to the dynamic process in the presence of active culture. Additionally, please consider discussing any major challenges or adjustments anticipated for this purpose.

Comments on the Quality of English Language

Some sentences might require paraphrasing. 

Reviewer 2 Report

Comments and Suggestions for Authors

 This work made minor modifications on the widely used correlation of van’t Riet for estimating the volumetric mass transfer rate kLa in single-use bioreactors of two scales (200 and 2000 L) and an industrial scale conventional stainless steel bioreactor (15000 L). Scale-up and scale down rules among the three scales are analyzed in terms of the mainly concerned performance of mass transfer rate. The results show that the modified van’t Riet correlation could provide a robust and reliable prediction of mass transfer coefficients on the three investigated scales for a wide range of stirrer tip speeds and aeration rates. The manuscript is generally well organized and the results can provide meaningful insights into the design and scale-up of bioreactors. Some minor questions are:

(1)    Please give the definition of the dimensionless mixing time.

(2) At constant stirring speed, with the increase of gas aeration rate, the state of gas dispersion in the stirred tank may vary. As the investigated conditions cover somewhat wide ranges, i.e., different scales, different impellers of combination of impeller, wide ranges of stirring speeds and aeration rates, are the gas well dispersed in all these cases?
